# Tools and Biomarkers for the Study of Retinal Ganglion Cell Degeneration

**DOI:** 10.3390/ijms23084287

**Published:** 2022-04-13

**Authors:** Ciriaco Corral-Domenge, Pedro de la Villa, Alicia Mansilla, Francisco Germain

**Affiliations:** 1Instituto Ramón y Cajal de Investigación Sanitaria (IRYCIS), 28034 Madrid, Spain; ciriaco.corral@salud.madrid.org (C.C.-D.); pedro.villa@uah.es (P.d.l.V.); 2Ophthalmology Service, Hospital Universitario Ramón y Cajal, 28034 Madrid, Spain; 3Department of Systems Biology, Universidad de Alcalá, 28871 Alcalá de Henares, Spain

**Keywords:** retinal ganglion cells, neurodegeneration, markers, apoptosis, glaucoma

## Abstract

The retina is part of the central nervous system, its analysis may provide an idea of the health and functionality, not only of the retina, but also of the entire central nervous system, as has been shown in Alzheimer’s or Parkinson’s diseases. Within the retina, the ganglion cells (RGC) are the neurons in charge of processing and sending light information to higher brain centers. Diverse insults and pathological states cause degeneration of RGC, leading to irreversible blindness or impaired vision. RGCs are the measurable endpoints in current research into experimental therapies and diagnosis in multiple ocular pathologies, like glaucoma. RGC subtype classifications are based on morphological, functional, genetical, and immunohistochemical aspects. Although great efforts are being made, there is still no classification accepted by consensus. Moreover, it has been observed that each RGC subtype has a different susceptibility to injury. Characterizing these subtypes together with cell death pathway identification will help to understand the degenerative process in the different injury and pathological models, and therefore prevent it. Here we review the known RGC subtypes, as well as the diagnostic techniques, probes, and biomarkers for programmed and unprogrammed cell death in RGC.

## 1. Introduction

The eye represents an ideal organ for non-invasive imaging. The transparency of its interior environment makes the eye an accessible ‘window’ to the central nervous system (CNS). Since the retina is part of the CNS, many of the results of its study can be extrapolated to different areas of the CNS. In addition, neurons and synaptic connections of the retina are arranged in a very organized manner and disposed of in perfectly differentiated layers. This very well-known organized structure can be accessed to carry out functional or structural studies easily. The retina is under study due to the increased prevalence of retinal pathologies and the need to improve diagnosis, but also the retina is established as a tool in the study of neurodegenerative alterations that go beyond the visual system, as seems to be the case with Alzheimer’s or Parkinson’s diseases [1,2,3].

Retinal ganglion cells (RGC) transmit light and visual information to the brain via the long axons that form the optic nerve. Specifically, the human retina contains ~1.5 million RGCs, and they are located mainly in the ganglion cell layer and some of them in the inner nuclear layer [4]. These RGCs do not have any discriminating optical properties, so fundus photography, confocal scanning laser ophthalmoscopy (cSLO), or laser polarimetry alone, would be inadequate to visualize them (for information on these techniques see below). This has led to the development of RGC markers to accurately visualize, diagnose, and monitor RGC degeneration in ophthalmological diseases, such as glaucoma [5,6,7,8]. RGCs are not homogeneous, series of types and subtypes of still indefinite number are described based on their light response, morphology, or gene expression criteria. The importance of specifically studying the different RGC subtypes is double. First, each RGC subtype is believed to transmit a specific type of visual information for later integration in the brain [9]. Second, each RGC subtype has a distinct susceptibility to disease and nociceptive stimuli [10].

Early markers for RGC degeneration are being developed, which will improve our understanding of the pathogenesis throughout the course of the disease, providing us with tools to measure the neuroprotective efficacy of therapeutic agents [6,7,11,12]. This is particularly important because in current clinical practice the tools available to quantify the degree of neuroprotection are not well established.

Here a review of the existing markers for different RGCs subtypes has been carried out, as well as the techniques and markers to study programmed cell death in vivo and in vitro, and finally the relevant markers of different degenerative cellular processes other than apoptosis have been reviewed. Its interest lies in the possibility of studying different RGC subtypes since they carry different information and have also shown a differential resistance to injury.

## 2. RGC Classification and Injury Sensitivity

In the first RGC classifications made in different animal species, both morphological and functional criteria were followed. The first systematic classifications used morphological criteria from the cat retina and differentiated among the alpha, beta, and delta subtypes [11]. However, in other species, this classification was difficult to maintain. In addition, with the development of new analysis techniques, it was observed that some of these subtypes were divided into others or that they shared aspects with others that were initially different. Currently, most reports classify RGCs (and neurons in general) according to a set of shared characteristics related to genetics, immunohistochemistry, physiology, and morphology (including the dendrite arbor, soma, and axonal projections) [12,13,14,15]. These characteristics define cell types and guide the analysis, but confirmation of RGC purity sometimes involves the consideration of other criteria, such as mosaic distribution. 

As the mouse is the main animal model in the study of retina degeneration, we are going to review RGCs classification in mice. The functional classification referred to here, considers 42 subtypes of mouse RGCs and its main guiding criterion is the type of response: light increment response or ON versus light decrements response or OFF and spike trains of brisk transient versus sustained (Table 1). In turn, these types have been subdivided according to other data (morphology of the dendritic arbor, size of the receptive field, being intrinsically photosensitive, vertical or horizontal preference axis, movement sensors, etc.), or even transcriptomic data [16] (Appendix A). With the application of these criteria, some of the classic subtypes RGCs, such as alpha cells, have had to be regrouped within other categories based on their response. Another important group is represented by the intrinsically photosensitive retinal ganglion cells (ipRGC). In 2000, the presence of melanopsin in a small percentage of retinal ganglion cells was found [17]. Since then, six subtypes of ipRGC (M 1–6) have been described. These subtypes of RGC present different responses, which is why they can be found in different sections according to the criteria used (Table 1 and Appendix A).

Currently, there is great controversy about the susceptibility of specific RGC subtypes to injury, with diverse vulnerabilities being observed even in the same injury model or different susceptibility in the same RGC subtype depending on the insult [10]. There is a significant difference between chronic models (bead occlusion) and acute models of injury (crushed optic node), and within an RGC subtype, differences are also observed depending on its modality, for example, ON alpha is more resistant than OFF alpha [18]. There is another subgroup, the ipRGCs, made up of six subvarieties (M1 to M6) with different functional responses and resistance to injury. In general, the ipRGC turn out to have good resistance to the majority of insults like optic nerve section [19] or NMDA excitotoxicity [20], but it is the RGC M1 that seems to be the most resistant to the transection of the optic nerve [21]. Some studies suggest that overall the alpha and ipRGC subtypes are the most resistant to injury [22]. In one of the first studies that analyzed the differential sensitivity of the RGC, it was observed that two subtypes were more resistant to axotomy than the others [23]. One of the subtypes was described as a ‘‘large RGC’’, resembling the ON alfa-like ganglion cell, which corresponds morphologically to M4 melanopsin ganglion cells of the mouse retina [24]. The other one was a new RGC subtype, the only survivor one year after axotomy, whose morphological description coincides with that of M1 ganglion cells. Also, there was a minority (*ca*. 1%) of surviving RGCs, which correspond to a different subtype. It cannot be ruled out that this subtype correlates with another melanopsinic cell, but with poorly detectable levels of melanopsin. 

There are several possible explanations for the better survival of M1 RGC. One is the close relationship to dopaminergic amacrine cells. Dopamine has been shown to be neuroprotective against glutamate-related neurotoxicity [25]. Thus, synaptic contacts between M1 RGC dendrites and dopaminergic amacrine cells processes [26] could provide support to the RGCs to prevent degeneration. Another possibility is that surviving RGCs may be supported by other cells located in the INL, through collateral axons that innervate the IPL [27], creating a new circuit to help M1 RGC after injury [21]. 

Apart from other routes, such as PTEN/mTOR pathway that provides protection against many pathologies including neurodegenerative diseases [28], and JAK/STAT [29], involved in cellular survival, proliferation, differentiation, and apoptosis [29], there is a third one, the neuropeptide pituitary adenylate cyclase-activating polypeptide (PACAP) pathway, which has a cytoprotective action in neurons and other cells [30]. Interestingly, PACAP is colocalized with the melanopsin-containing ganglion cells [31]. Moreover, it is known that proapoptotic signals arrive through the gap junction but it has been postulated that melanopsin RGCs express Cx30.2 [21], which has the lowest single-channel conductance among all members of the connexin family [32]. In addition, they are coupled to amacrine cells rather than ganglion cells. The joint effect of these two factors could prevent the transfer of apoptotic signals to melanopsinic RGC [21].

**Table 1 ijms-23-04287-t001:** General classification of RGC subtypes. RGC subtypes classification based on definitions from Gregory W. Schwartz’s research group at Northwestern University (USA) [16,33,34] Abbreviations for cell types: M1–M6: melanopsinic RGC, OS: orientation-selective, DS: direction-selective, s: sustained, tr.: transient, Me: medium, Sm: small, Lg: large, RF: receptive field, h: horizontal, v: vertical, SbC: suppressed by contrast, b: bursty, HD: High definition, UHD: ultra-high-definition, LED: Local edge detector, EW: Eyewire.

Main Feature	ON/OFF Feature	Subtypes
**Sustained**	ON	ON alpha, Pix ON, M2, M1
	OFF	OFF s alpha, OFF s med, OFF s EW1 no, OFF s EW3o, OFFhOS, OFFvOS
**Transient**	ON	M6, ON tr MeRF, ON tr SmRF, ON tr EW6t
	OFF	OFF tr alpha, OFF tr MeRF, OFF tr SmRF
**Orientation selective**	ON	ONhOS SmRF, ONvOS SmRF, ONhOS LgRF, ONvOS LgRF
**Direction selective**	ON	ONDS s (3 subtypes), ONDS tr (1 subtypes)
	ON-OFF	OODS (4 subtypes)
**ON-OFF Small RF**	ON-OFF	HD1, HD2, UHD, LED, F-mini ON, F-mini OFF
**SbC/Others**		ON delayed, ON bursty, bSbC, sSbC EW27, sSbC EW28, ON Sm OFF Lg, Motion sensor

## 3. RGC Labeling

Imaging of RGCs remains challenging due to the transparent nature of the retina. In order to observe such cells differentially, specific RGC labeling techniques can be combined with retinal imaging. In clinical practice, contrast agents are commonly used to enrich the structural and functional information of positron emission tomography, computed tomography, and magnetic resonance imaging, or simply in angiography by injecting fluorescein and indocyanine green. The incorporation of contrast agents helps in the accurate diagnosis, detection, and monitoring of degeneration. Experimental techniques for labeling neurons in vivo use retrograde labels, transgenic templates, or electroporation among others [35]. In addition, certain probes allow imaging of diseased cells in vivo, like CapQ and Annexin-A5, which are valuable tools to study pathological mechanisms of neurological and retinal degeneration [36,37]. 

There are also other techniques to label cells like electroporation, which involves applying a voltage across a membrane to introduce ectopic genes or contrast agents into cells in vivo [35]. The eye presents a certain ease in administering contrast agents, since a subretinal or intravitreal injection, depending on the location of the retinal cells, can reach the proposed target [38]. Unfortunately, this technique has the potential ability to irreversibly damage RGCs, as they do not regenerate. However, the optimal pulse required, in such a small tissue, is much less intense [39], and the damage that can be produced is smaller. Another disadvantage is its lack of specificity for a particular cell type [40]. The RGC is found to share the same retinal layer with the displaced amacrine cells, so both types of cells could be affected by electroporation.

### 3.1. Dyes for RGC Labeling

Retrograde labeling involves the direct application of lipophilic neuronal dyes to the visual pathway. It is a well-established tool in histological studies for RGC quantification, although it is not applicable for in vivo studies in humans. Different types of fluorescent dyes, which are actively transported within cells, have been used [41] such as carbocyanin dyes: DiAsp (4-[4-(didecylamino)styryl]-N-methylpyridinium iodide) [42], DiI (1,1′-dioctadecyl-3,3,3′,3′-tetramethylindocarbocyanine perchlorate) [43], DiO (3,3′-dioctadecyloxacarbocyanine perchlorate) [44], DTMR (dextran tetramethyl rhodamine) [45], and Fluorogold (2-hydroxystilbene-4,4′-dicarboxamidine bis(methasulfonate) [46]. In studies using retrograde transport, the tracer is applied to a fiber or innervation target. Once the tracer is incorporated into the cell’s axons by endocytosis, it is transported towards the cell body. In contrast, in anterograde transport, the uptake mechanism involves the cell body and/or its dendrites and later the tracer is transported along the microtubule system to the distal synaptic terminals. [47]. They have been used both ex vivo and in vivo [7,48,49]. After injecting them into higher centers of the visual pathway, they travel retrograde down the RGC axon until they reach the cell body located in the retina [50,51,52]. An example is the direct injection of DiI into the superior colliculi of the rat, which allows the analysis of RGC lost after increasing intraocular pressure. Imaging can be performed by a confocal scanning laser ophthalmoscope (cSLO). In addition to injecting in the superior colliculus [48,53,54], tracers have been injected into the lateral geniculate nucleus [7] and the optic nerve [41], achieving good results.

Retrograde tracers have also been used to assess axonal structural degeneration, which involves impaired axonal transport. The advantages and disadvantages of using other labels such as fluorescent horseradish peroxidase (HRP), rhodamine-B-isothiocyanate (RITC), or cholera toxin B (CTB) for axonal degeneration and RGC loss in glaucomatous optic neuropathy were analyzed elsewhere [50]. 

Sometimes retrograde cell labeling is not complete. The cause seems to be the incomplete uptake of the dye by the RGC, or that not all the central targets of the RGC were marked. Even so, a high efficacy label of rodent RGC has been demonstrated with these methods. Labeling is estimated to vary between 96% and 100% RGC after application in the superior colliculus and direct injection in the severed optic nerve stump, respectively [48]. However, after the death of some RGC, microglial cells and macrophages migrate to the area and ingest fluorescent cell debris, which can generate false positives, only distinguishable from neurons by their shape and size [49,51].

Anterograde labeling is also possible with some of these dyes, for example with cholera toxin B [52]. After intravitreal injection of this neuronal biomarker in mice, RGCs were efficiently marked [53] and by anterograde transport, the entire visual pathway is marked as well [54]. This labeling did not produce cell toxicity, but amacrine cells are also labeled [52,54].

### 3.2. Immunostaining: Specific Antibodies against RGC Subtypes

One of the best ways to label RGC and distinguish subtypes is protein detection by immunohistochemistry in whole-mount or sections from postmortem retinas. The main advantage of this classical technique is the possibility of simultaneously marking several proteins and seeing their interrelation. 

There are few markers that reliably identify RGC: the brain-specific homeobox/POU domain protein 3A (Brn3a) [55], the RNA-binding protein with multiple splicing (RBPMS) [56], gamma-synuclein [57] or the membrane protein CD90 also known as Thy 1.2 [58]. Besides the use in postmortem retinas, antibodies recognizing these proteins have been used for labeling RGC in vivo [7,59] or ex vivo [56]. 

Several ganglion cell subtypes are differentiated by the exclusive expression of some proteins, which allows specific subtypes labeling. For more information on subtype-specific markers see Table 2.

### 3.3. Transgenic Models

Transgenic models have been widely used in many fields of scientific research to control the expression of specific genes and study the function of certain proteins. Under the control of neuronal-specific promoters, the expression of fluorescent proteins, such as green fluorescent protein, can act as a cell-specific marker for RGC. Transgenic mice expressing fluorescent molecules observed by spectral-domain OCT and cSLO have made it possible to measure changes in RGC density and layer thickness over time, for example after optic nerve injury [75,76]. Images of transgenic mice expressing cyan fluorescent protein via the Thy1 promoter were taken using cSLO to enhance contrast in labeled different neuronal subsets [5]. To see more transgenic mice lines to study RGC subtypes see Appendix A.

Disadvantages of using fluorescent transgenic lines are the low specificity, since it may mark displaced amacrine cells, and that after phagocytosis of apoptotic cells by microglia, these phagocytic cells also appeared marked [51], or that in some studies the transgenic labeling has been transitory, not giving enough time to carry out longitudinal studies.

## 4. Neurodegeneration and Death of Retinal Ganglion Cells

Neurodegenerative diseases, such as glaucoma, cause the progressive loss of RGC via apoptosis. The existence of a window period between the injury and the onset of RGC death allows some treatments to have a chance. However, there are several problems, such as the fact that current diagnostic techniques require a functional loss that can reach 25–40% of the RGC before a proper diagnosis is made [6,77]. This implies carrying out serial measurements for 2 to 8 years, depending on the course of the disease, before being able to detect these changes [78]. In addition, it also requires the collaboration of the patient, which cannot always be obtained [79,80]. Thus, by the time the treatment is applied, it is no longer as effective, and the patient may become blind. Therefore, early diagnosis is an absolute necessity. 

Several ocular and extraocular neurodegenerative diseases share the common characteristic of early and pathological death of retinal cells. Getting an early diagnosis could slow down, or even stop, disease processes before they cause significant damage. Detection of apoptosis in retinal cells seems a plausible means of achieving this goal. Different strategies and technologies have been designed for this, some of which are still in the process of being developed.

Apoptosis is a key component in development and aging, but it is also the mechanism of death followed in various neurodegenerative and autoimmune diseases [81,82,83]. Previously, many of these diseases were associated with other causes. Glaucoma, for example, was attributed to elevated intraocular pressure. However, there were cases of glaucoma with low intraocular pressure. Even if RGC apoptosis is now established as the main cause of glaucoma, that doesn’t mean high intraocular pressure is not related to the disease. High pressure is associated with altered axonal traffic, thus reducing the nutritional supply and trophic factors, which worsened in situations of metabolic stress and ultimately kill RGCs by apoptosis.

In vivo detection techniques of apoptosis must be able to analyze the presence of markers in the cell membrane, like the activation of caspases and the exposure of phosphatidylserine [84]. The early identification of these markers through a non-invasive method is a key objective in biomedical research in order to perform precise diagnoses and apply effective treatments. The basic imaging techniques of the retina that, together with specific labeling of RGC, allow its assessment, would be optical coherence tomography and confocal scanning laser ophthalmoscopy (review below). For a more exhaustive comparison of the different diagnostic imaging techniques of the retina see a complete revision elsewhere [35].

In addition, many of these diseases throughout their degenerative process presented excitotoxicity, oxidative stress, mitochondrial dysfunction, or misfolded protein aggregation [85]. All these events left hallmarks in the cell that could be observed to perform a better diagnosis of the degeneration process. 

### 4.1. Retinal Images Techniques

Optical coherence tomography (OCT) can be temporal or spectral domain. This technique generates two-dimensional cross-sectional images from the optical backscattering of light and the time delay of the echo. If enough scans are available, it can generate three-dimensional images. In addition, its axial resolution is very high (1–15 μm) [86,87]. However, weak backscatter and low-contrast cell borders prevent direct visualization of individual RGC. The fundamental value of OCT is to provide high-resolution images of the thickness of the retinal nerve fiber layer (RNFL) [75,88,89,90]. Its performance has been improved by applying a Doppler development [91]; as well as with polarization-sensitive OCT, since ocular structures are capable of altering the polarization state of light, adding tissue-specific contrast to images, and allowing RGC axon densities to be measured [92]. Similarly, long-wavelength OCT, which reduces scatter [84], and swept-source OCT, which uses a tunable long wavelength can further increase resolution, represent significant improvements.

Confocal scanning laser ophthalmoscopy (cSLO) was established in 1980 [93], and is the most widely used retinal imaging technique. Its remarkable ability to adapt has been key to its survival as a diagnostic technique in glaucoma, providing high-resolution images with which subtle changes in the retina can be detected [94].

The cSLO methodology is based on the confocal microscopy technique, in which a laser scans the retina, and thanks to the pinhole, only light coming from certain depths is detected. In addition, as it is a narrow beam of light, no scattered light is produced; ensuring that only light from the desired focal plane is detected. In this way, a high lateral resolution is achieved, which allows the production of topographic images [93]. However, the axial resolution of conventional cSLO is poor. The key to improving the functionality of cSLO is the use of fluorescent markers. So, several research groups have used this combination to detect apoptotic RGCs in vivo using endogenous or exogenous markers and to be able to longitudinally monitor rodent RGCs in vivo [6,11,48,67,87,88]. In addition, the possibility of using various types of filters and laser wavelengths would allow double or multiple label detections. Limitations on its use only include pupil diameter and ocular opacities. However, the future of cSLO lies in adaptive optics (AO), in the ability to alter the scan amplitude speed, and in modifying the size of the pinhole.

One of these technical improvements in the ophthalmological field has been achieved through adaptive optics, since it allowed reducing optical aberrations [95]. Intrinsic optical aberrations were detected by analyzing the wavefront of light coming from the eye, and were corrected by electro-actuated deformable mirrors [96]. The main technical advantages of AO are improved lateral and axial resolution of retinal images, detection of smaller dots, and improved sensitivity to weak reflections. In combination with cSLO and OCT, it can improve fine detail resolution in vivo [95]. The association of AO with OCT increases the lateral resolution by five times compared to standard OCT, allowing direct visualization of individual cells without the need for an exogenous marker. This resolution allows individual nerve fiber bundles to be observed in humans in vivo [97,98]. Association with cSLO allows high-resolution in vivo visualization of fluorescently labeled rodent capillaries [99]; direct observation of cone-type photoreceptors through their intrinsic reflectance [100], and better resolution to analyze RNFL. The AO-SLO association was more accurate than the OCT [101]. All the above indicates that the future of cSLO lies in AO development. Although, in parallel, research is being carried out on small-aperture fast-scanning cSLO [102]. On the other hand, fluorescent labelers have an important role in further improving this technique.

### 4.2. Apoptosis Detection Techniques

The Detection of Retinal Apoptotic Cells (DARC) is a recent methodology that, through real-time non-invasive imaging technique applied to the detection of apoptotic RGC cells in vivo, has the potential to identify diseases in their early stages [6]. It consists of the injection of intravenous annexin-5 marked with fluorescence (ANX776). Annexin 5 has a high calcium-dependent affinity for negatively charged phosphatidylserine [103]. It has been observed that, during the early phase of apoptosis, phosphatildylserine is externalized in the outer membrane of neurons. [104]. Binding of phosphatidylserine to annexin 5 in the plasma membranes of apoptotic cells is detected by cSLO retinal examination. This examination allows counting the number of fluorescent spots representing each simple apoptotic RGC bound to annexin-5 to be calculated. During the cellular stress of early apoptosis, phosphatidyserine, normally intracellular, translocate to the outer plasma membrane, and is exposed to the outside of the cell, signaling it to be removed by phagocytic cells [105]. As this is one of the initial steps of the apoptotic cascade, it constitutes a much earlier marker than others, such as DNA fragmentation detected by the terminal labeling of deoxynucleotidyl transferase dUTP Nick (TUNEL) [6,36,96,98].

Detection of annexin-5 labeling by confocal scanning ophthalmoscopy (cSLO) focused on the RGC layer [106] allows for obtaining high-contrast fluorescent images that span between 35° and 55° of the retinal field. On the other hand, the ability of annexin 5 to cross the blood-brain barrier [107] allows the assessment of diseases in other areas of the nervous system, in addition to the retina.

This technique has proven useful in the investigation of neuroprotective therapeutic agents in glaucoma models, as well as in the relationship between glaucoma and Alzheimer’s disease [6,108,109] or the correlation between the number of apoptotic cells and axonal loss in RGC [110]. It has also served to establish the relationship between increased intraocular pressure and RGC apoptosis [108], and to verify in an intraocular hypertension model the reduction of apoptosis in vivo by the use of coenzyme Q10 [111]. Likewise, in the retina of diabetic mice, an increase in DARC counts was observed before vascular changes were perceived in the eye [112]; in a model of blue light exposure in rats this technique served to determine photoreceptors loss [113]; it made possible to verify the neuroprotective effect of brimonidine in glaucoma and the prevention of the formation of amyloid plaques [114], especially useful for early visualization of Alzheimer’s disease; it was used to determine the protective effect of modulating glutamatergic excitotoxicity [115]; or to demonstrate the therapeutic capacity of 2-Cl-IB-MECA to reduce apoptosis in vivo after partial transection of the optic nerve [116]; it was essential to detect the regeneration of injured axons after placing Schwann cells on the damaged optic nerve sheath [117]; in the same model, topical recombinant human nerve growth factor (rh-NGF) was found to be able to decrease apoptosis [118] by using this technique.

Especially interesting is the fact that this technique allows, through the retina, to observe the evolution of other diseases whose main symptoms affect other systems. Thus, the analysis of amyloid plaques in the retina and their relationship with apoptosis have shown a dose- and time-dependent relationship, so that, by preventing their formation or increasing their elimination, survival was increased [119]. In general, a close relationship has been observed between the retina and diseases in other parts of the nervous system, such as Alzheimer’s [120,121] or Parkinson’s diseases [122]. Therefore, the study of the retina seems to be a good way to control the evolution of these other diseases of the nervous system.

### 4.3. Caspase Activation Detection

Caspases are essential endoproteases in apoptotic and inflammatory processes. Caspase activation occurs through extrinsic or intrinsic signals. The extrinsic pathway is triggered by ligands that bind to extracellular death receptors, while the intrinsic pathway responds to intracellular stress signals such as hypoxia, DNA damage, reactive oxygen species, accumulation of misfolded proteins, and mitochondrial damage. Regardless of the trigger, the cascade begins with the activation of “starter” caspases capable of cleaving and activating “executer” caspases, which cleave DNA leading to cell death. Detection of these caspases is a reliable indicator of apoptosis. On the other hand, some data suggest that different types of neurons use different death messages. Thus, in the retina, caspase-1 plays an important role in photoreceptor death, while caspase-3 is important in the inner nuclear layer, and caspase-2 is the main caspase involved in neuron death in the retinal ganglion cell layer [123].

Another option to detect caspase activity is the use of apoptotic probes (CapQ) activated by caspases. These penetrate cells and mark those that are in apoptosis [36]. This technology consists of a cell-penetrating peptide conjugated to an effector caspase recognition sequence, joint to a pair of fluorophores. This set is activated by effector caspases in apoptotic cells, and its fluorescence is detected by cSLO. After intravitreal injection of the TcapQ488 probe, RGCs showing apoptosis in vivo were detected using cSLO in mouse retinal degeneration models [37]. However, this probe had minimal toxicity capable of activating the probe, even in the eyes of wild-type rodents, and therefore causing apoptosis.

The use of caspase inhibitors with fluorescence (FLIVO, fluorescence in vivo) allows the visualization of apoptosis in vivo and in vitro. These tracers injected into the circulation selectively accumulate in apoptotic cells. Being able to cross the blood-brain barrier, they can be used in the study of brain and eye neurodegenerative diseases, selectively targeting cells that undergo caspase-dependent apoptosis [124]. These methodologies have been used in animal models (in vitro and in vivo) and in the clinic to monitor the activity of Diabetic Retinopathy [125], glaucoma [126], and retinitis pigmentosa [127], blue light-induced retinal damage, and AMD [128].

Luciferins are bioluminescent molecules that when activated by luciferases release energy by emitting light. Z-DEVD-aminoluciferin is a luciferin modified to be activated by specific caspases, which allows it to detect the activity of these caspases in vitro [129]. In this way, it serves as a marker of apoptosis. Apoptosis has also been detected in vivo in mouse models of tumor xenografts [130].

### 4.4. Detection of Changes in the Apoptotic Membrane

Certain imaging techniques to detect apoptotic cells also use low molecular weight (300 to 700 Da) amphipathic molecules that selectively cross the apoptotic plasma membrane and accumulate in its cytoplasm as the Aposense family of compounds [131,132]. The anchoring is made to the hydrophobic (lipid) region of the cell membrane, passing into the cell interior, unlike what happens in non-apoptotic cells, in which the hydrophilic region blocks their entry into the cytoplasm. Examples of these compounds are those containing the dansyl group, like N,N′-didansyl-L-cystine, NST-732, and NST-729; or those containing an alkyl-malonate molecule, such as ML-9 and ML-10 [131].In apoptosis, the accumulation of these molecules in the cytoplasm, exposure to phosphatidylserine, activation of caspases, and loss of mitochondrial membrane potentials have been observed. These compounds may be intrinsically fluorescent or may be labeled with a radioactive moiety. These molecules have demonstrated their usefulness in the field of experimentation in models of Alzheimer’s disease, amyotrophic lateral sclerosis [133], melanomas [134], chemotherapy-induced enteropathy [135], and models of reperfusion-induced damage [132]. In clinical practice, a radiolabeled version of ML-10 has been used to monitor the response of brain metastases to radiotherapy [136]. Although these molecules can cross the blood-brain barrier, and therefore could be used in neurodegenerative conditions such as Alzheimer’s, Parkinson’s, and glaucoma, their toxicity could be a problem.

Glaucoma ocular biomarkers are endogenous biochemical, physiological, and anatomical indicators associated with specific pathological states [137,138]. They provide an objective measure to detect the disease early and monitor therapeutic efficacy. The optimal biomarker must be specific, sensitive, and reproducible, as well as inexpensive and non-invasive. A glaucoma biomarker should indicate the rate of RGC loss and the number of remaining or apoptotic RGCs with high sensitivity. Recent advances in fluorescent technology have improved the ability to identify individual RGCs undergoing apoptosis. These specific markings will be valuable both in experimental models and in the clinic.

## 5. Non-Apoptotic Biomarkers

In addition to the typical markers and techniques to show the apoptosis process, there are other proteins that change their expression, their chemical state, or their location because of the degeneration or specific pathological insult [138]. These biomarkers may reflect the activation of cellular responses previously or concomitant with apoptosis or even totally independent pathways.

### 5.1. Membrane Markers

Caveolins are plasma membrane proteins present in specific cell membrane structures called caveolae. These membrane specializations are lipid rafts that are involved in endocytosis and important cell signaling processes. There are three types of caveolins, Cav1, Cav2, and Cav3. Both Cav1 and Cav2 are expressed in the retina [139], including ganglion cells and vascular tissue. Mutations in the gene region of Cav1 and Cav2 have been identified as glaucoma-linked variants, and loss of Cav1 in mice leads to changes in retinal vessel morphology and electrophysiological decrease function of RGC [140,141]. Therefore, the loss of function of caveolin 1 has been associated with the degeneration of ganglion cells linked with pathology.

### 5.2. Oxidative Stress Markers

Reactive oxygen species (ROS) are chemical molecules, radicals, and non-radicals, that act as oxidizing agents and/or are easily converted to radicals. ROS are the cause of oxidative stress, which involves lipids, proteins, enzymes, carbohydrates, and nucleic acids damage, inducing cell death by nucleic acid fragmentation and lipid peroxidation. Under normal conditions there is a balance between the production of ROS and the antioxidant enzymes that counteract them, the pathological mechanism is triggered when this balance is broken either by an increase in ROS or by a decrease in antioxidants [142]. The retina, due to its high metabolic activity and reduced regeneration, is particularly susceptible to the toxic effects of ROS [143]. 

Oxidative stress, related to RGC degeneration, has been described in glaucoma, diabetic retinopathy, or retinal ischemia-reperfusion injury [143,144,145]. In glaucoma, there are two theories about RGC degeneration, the vascular theory, and the mechanical theory, both of which can be explained by oxidative stress. In the vascular theory, the compromise of the retinal vessels causes ischemia and thus an accumulation of ROS. In the mechanical theory, the excessive formation of ROS is due to the lack of neurotrophins that arrive retrograde through the axon, this neurotrophin flow is interrupted due to the high IOP [143,146]. Fluorescence-based probes are the easiest way to monitor the concentrations and location of these often very short-lived ROS. These probes are compounds that fluoresce when oxidized in the presence of specific ROS, they have high levels of sensitivity and the ability to be used for temporal and spatial sampling for in vivo or in vitro imaging applications [147]. Some examples are 2–7 dichlorofluorescein diacetate (DCFDA) which produces green fluorescence when is oxidized by hydrogen peroxide or Dihydroethidium (DHE) which oxidizes in the presence of superoxide anion, generating 2-hydroxyethidium, a red fluorescent compound. DCFDA, DHE, and other fluorogenic compounds have been used to study oxidative stress in several mouse models of retinal pathology; diabetic retinopathy [136], and optic neuropathy [148], among others.

It is possible to measure oxidative stress indirectly by measuring cellular response to ROS. One of the main molecules in the oxidative stress response is the transcription factor (erythroid-derived 2)-like 2 (Nrf2). Under normal conditions, Nrf2 is found in the cytoplasm, but when ROS levels increase, Nrf2 translocates to the nucleus and induces the expression of a group of antioxidant enzymes such as superoxide dismutase (SOD) or catalase [149,150]. This nuclear translocation can be studied by immunohistochemistry in retinal sections. In an experimental model of retinal ischemia-reperfusion, Nrf2 knockout mice exhibited a much greater loss of neuronal cells in the ganglion cell layer than wild-type mice [151].

Iron is crucial for cellular metabolism, but free iron catalyzes the conversion of hydrogen peroxide to the hydroxyl radical, the most reactive of ROS. Iron-dependent oxidative stress can cause cell death, specifically, ferroptosis. Photoreceptors and retinal pigment epithelium are particularly dependent on iron metabolism and ferroptosis has been associated with the pathogenesis of age-related macular degeneration [152]. A role for iron-induced oxidative stress in RGCs in glaucoma has also been suggested. Iron-related proteins transferrin, ceruloplasmin, and ferritin serve as iron oxidative stress markers and were shown to be upregulated in a monkey model of glaucoma, and human postmortem glaucomatous eyes [153].

### 5.3. Mitochondrial Dysfunction Markers

RGCs have a tremendous energy requirement from mitochondrial function to relay visual information to the brain. Mitochondrial dysfunction has a causative role in the pathogenesis of optic neuropathies such as Leber’s hereditary optic neuropathy and autosomal dominant optic atrophy [154,155], and plays a significant role in the neurodegenerative cascade of RGCs in glaucoma [156]. On the other hand, non-genetic mitochondrial dysfunction is associated with increased oxidative stress, which directly alters the structure of mitochondrial macromolecules and enzymes and damages mitochondrial DNA which results in irreparable mutations. Mitochondrial DNA is particularly susceptible to oxidative damage due to relatively poor DNA repair mechanisms and an absence of protective histones and DNA-binding proteins. Accumulation of general mitochondrial damage causes cytochrome c release and concomitant activation of caspases that commits the cell to apoptosis [154].

Flavoproteins and the nicotinamide adenine dinucleotide (NADH) and nicotinamide adenine dinucleotide phosphate (NADPH) coenzymes, are essential components of normal mitochondria and participate in the redox reactions that are fundamental to cellular respiration [19,20]. Oxidized, but not reduced, flavoproteins display autofluorescence on the contrary NADH and NADPH fluoresce when reduced but with different excitation-emission wavelengths from flavoproteins. The ratio of oxidized to reduced forms varies in environments of oxidative stress making these autofluorescence proteins good markers of mitochondrial dysfunction by oxidative stress, and so have been used in vitro and in vivo in retinal pathologies [157].

Markers of mitochondrial dysfunction independent of oxidation are mitochondrial membrane potential dyes and the translocation of Apoptosis-inducing factor (AIF). Membrane-potential-dependent dyes, such as the commercially available MitoTrackers, label mitochondria permanently within live cells utilizing the mitochondrial membrane potential. Mitochondrial mass, structure, and membrane potential are easily followed with these probes [158,159]. AIF is a protein strictly confined to mitochondria in normal conditions and thus colocalizes with heat shock protein 60. In mitochondrial damage conditions or induction of apoptosis, AIF translocates to the nucleus and induces chromatin condensation and large-scale DNA fragmentation, independently of caspases [160]. AIF translocation and MitoTrackers’ reduction have been observed in a cellular model of RGCs subjected to hypoxia [161] and in RGCs from a rat glaucoma model [156].

### 5.4. Endoplasmic Reticulum (ER) Stress Markers

Disease-related cellular dysfunctions result in ER stress, commonly followed by an accumulation of unfolded proteins in the ER lumen and the subsequent activation of the unfolded protein response (UPR). UPR consists of a shutoff of protein translation and switching-on specific transcription factors to control genes that function to reduce unfolded proteins and restore ER homeostasis. If this vital process fails, the cell will be signaled to enter apoptosis [162]. 

UPR consists of three pathways defined by three ER proteins: activating transcription factor 6 (ATF6) inositol requiring protein 1 (IRE1), and protein kinase RNA (PKR)-like ER kinase (PERK). The luminal domains of these ER transmembrane proteins are sensors of protein misfolding whereas cytosolic regions interact with transcriptional and translational machinery to resolve protein folding burden. Under basal conditions, the luminal domains of these UPR sensors are bound by a chaperone binding immunoglobulin protein (BiP) and rendered inactive. In times of ER stress, BiP is titrated away, resulting in UPR activation. Unfolded proteins can also bind directly to IRE1 and PERK, resulting in dimerization, oligomerization, and ultimately activation. The three UPR initiation pathways converge in the activation of a series of transcription factors, among which the X-box binding protein 1 (XBP1) or C/EBP homologous protein (CHOP) stand out [163]. When ER stress occurs with high intensity, or is prolonged, homeostasis is not restored, and apoptosis is induced. CHOP plays an important role in ER stress-induced apoptosis in neurons [162,164].

It has been shown that the expression of ER stress markers (e.g., BiP, CHOP, or phospho-PERK) were significantly elevated in a rat model of chronic glaucoma [165] and in the retina of diabetic rats [166]. It has been shown that CHOP absence protects against RGC loss after ischemia/reperfusion retina injury [167].

### 5.5. Autophagy Markers

Autophagy is the digestion of small portions of the cytoplasm through the formation of double-membrane vesicles, whose content is degraded in the lysosomes. At the basal level, it occurs in practically all cells and it is generally considered a cytoprotective mechanism, maintaining homeostasis under starvation conditions and removing defective proteins, damaged organelles, and disease-causing pathogens; however, autophagy can also be harmful [168] inducing autophagy-dependent cell death. These antagonistic effects of autophagy have also been observed in the retina, since in vivo experimental studies of neurodegeneration (including glaucoma) pointed out that autophagy has neuroprotective effects on RGCs [169,170] but on the other hand, some studies have shown that autophagy might have unfavorable effects, as it promotes cell death in glaucomatous conditions [171]. Thus, in the case of autophagy, it is important to study time-dependent autophagic flux, because what begins as a protection mechanism ends up being deadly for the RGCs [172]. 

The microtubule-associated protein light chain 3 (LC3) II, is largely used as an autophagy marker. LC3 II is generated by the conjugation of cytosolic LC3 I to phosphatidylethanolamine on the membrane of nascent autophagosomes. Although it has been described in situations where the formation of autophagosomes is compromised and LC3 II accumulates non-specifically [173], the quantification of LC3-positive puncta is considered a gold-standard assay for assessing the numbers of autophagosomes and therefore autophagy levels. Accumulation of LC3 II in RGC has been shown between 6 and 24 h after a transient IOP increase in rats [174]. 

The protein p62, also called sequestosome 1, can interact with ubiquitinated proteins and serve to present them to the autophagic machinery to enable their degradation in the lysosome. p62 is itself degraded by autophagy, accumulates when autophagy is inhibited, and decreased levels are observed when autophagy is induced, so p62 has been established as a marker to study autophagic flux [175]. An accumulation of p62 has been reported in RGC axons following chronic IOP elevation [176]. To establish whether the increase in p62 reflects an increase or an impairment of autophagy flux it is necessary to study other autophagy markers such as beclin-1. One of the initial steps in the assembly of autophagosomes is the recruitment and activation of beclin-1. Also, upregulation of beclin-1 and LC3 II in RGC was reported following ON transection, a model mimicking the impairment of retrograde axonal transport of neurotrophins occurring in glaucoma [170]. To define autophagic flux it is also very useful to evaluate the levels of proteins that are part of the autophagosome formation process, these proteins, called atg proteins, like-atg5 atg4 or atg12 have been used successfully to assess the levels of autophagy in different models of glaucoma [175].

### 5.6. Necroptosis Markers

Necroptosis is defined as a form of regulated necrosis, it is caspase-independent and is characterized by a loss of cell membrane integrity and cytoplasm swelling, that results in inflammatory responses. Necroptosis can be triggered by specific death receptors followed by the concomitant activation of receptor-interacting protein kinase 1 (RIP1) and receptor-interacting protein kinase 3 (RIP3) that form a microfilament-like complex called the necrosome, that promotes the activation of mixed-lineage kinase domain-like protein (MLKL). There is evidence that both ischemia in vivo and hypoxia in vitro can induce necroptosis of RGCs [177]. RIP3 expression in the ganglion cell layer and the inner nuclear layer was increased after ischemia/reperfusion in mouse retinas [176].

In conclusion, the current development of diagnostic tools, probes, and biomarkers for the retina, makes it possible not only to distinguish retinal ganglion cell subtypes but also to detect degeneration pathways at different stages (Figure 1). It has been seen that the early detection of retinal pathologies can advance the onset of treatment, preventing blindness from occurring. In addition, the diagnostic implications that these techniques and markers have in nervous system pathologies besides the retina, such as Alzheimer’s or Parkinson’s, are promoting their development exponentially. However, there are currently some limitations, such as the complexity of RGC classification, which must consider morphological and physiological aspects, or the fact that the repertoire of markers is limited and that some techniques and markers can only be performed in vitro or in animal models. Nevertheless, the need to improve clinical diagnosis in humans is achieving great advances in this field.

## Figures and Tables

**Figure 1 ijms-23-04287-f001:**
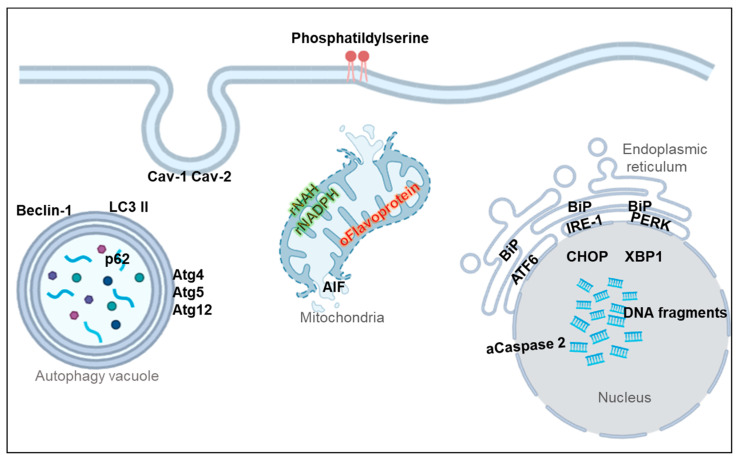
RGC degeneration biomarkers. Picture represents a cell and its organelles; markers are indicated in their preferred location. There are markers directly related to apoptosis such as phosphatidylserine (detected by Anexin V binding), activated caspase 2, or DNA fragmentation (detected by TUNEL, terminal labeling of deoxynucleotidyl transferase dUTP Nick). Mitochondrial dysfunction can be detected by measuring autofluorescence of reduced nicotinamide adenine dinucleotide (NADH) and nicotinamide adenine dinucleotide phosphate (NADPH) or oxidized Flavoproteins or by detecting AIF (apoptosis-inducing factor) release from the mitochondria to the nucleus. RGC degeneration may cause unfolded protein response and endoplasmic reticulum stress, which can be detected by measuring levels or changes in the localization of activating transcription factor 6 (ATF6), inositol requiring protein 1 (IRE1), the protein kinase RNA (PKR)-like ER kinase (PERK), the chaperone binding immunoglobulin protein (BiP) or the transcription factors X-box binding protein 1 (XBP1) and C/EBP homologous protein (CHOP). Changes in autophagy markers are also related to degeneration, markers such as becin-1 or autophagy proteins (Atg4, atg5, or Atg12), the autophagy substrate p62, and the microtubule-associated protein light chain 3 (LC3) II, are largely used as autophagy markers for RGCs. Totally unrelated to apoptosis are the levels of caveolins (Cav1 Cav2) in caveole membrane structures.

**Table 2 ijms-23-04287-t002:** Antibodies for RGC types and subtypes. The basic staining patterns and relevant characteristics are also provided where possible. Abbreviations: NeuN: reported synonym of the human protein ‘RNA binding fox-1 homolog 3′, encoded by the gene RBFOX3, CART: cocaine- and amphetamine-regulated transcript, Mmp17: matrix metalloprotease 17, Cdh6: cadherin 6, Col25a1: collagen 25a1, SMI 32: neurofilaments, SPP1 osteopontin, PV: Paralbumin, Brn3b: POU domain class 4 transcription factor 2, Brn3a: POU domain class 4 transcription factor 1, Brn3c: POU domain class 4 transcription factor 3, Foxp1/1: Forkhead box protein P1/1, A/T: Anterior/Temporal, N: Nasal, S: superior, I: inferior, ooDSGC: ON-OFF Directionally Selective Ganglion Cells, R-RGC: RGC labeled in the Rbp4-Cre mouse line, DRD4 Dopamine receptor D4, MMP17 Matrix metalloproteinase-17, SACs Starburst amacrine cells, F-RGC: F-mini/midi RGC, F-mini/midi: FOXP2 Positive mini/midi RGC, M1–M6: melanopsinic RGC, S-BGC, M-BGC, and B-BGC: Small Medium and Big Bistratified RGC.

Antibody	Cell Type	Reference
**Anti-NeuN**	RGC and amacrine nucleus	[60,61,62,63]
**Anti- cocaine- and amphetamine-regulated transcript (CART)**	Superior Inferior and posterior ooDSGC subtypes, does not stain A/T-ooDSGC. Purkinje+ RGC R-RGC	[64,65,66,67]
**Anti-matrix metalloprotease 17 (Mmp17)**	≤5% of all RGCs, some SACs 70% of MMP17 + RGCs are DRD4 + (subtype of N-ooDSGC) >90% of DRD4 + are Mmp17 +	[64]
**Anti-cadherin 6 (Cdh6)**	Marks both I-ooDSGC and S-ooDSGC in similar proportion. Marks SACs	[64]
**Anti-collagen 25a1 (Col25a1)**	Marks both I-ooDSGC and S-ooDSGC. Marks SACs	[64]
**Anti-SMI 32 (neurofilaments)**	Marks Alpha RGCs	[68]
**Anti- osteopontin (SPP1)**	Marks Alpha RGCs	[68]
**Anti-Brn3b/Po4f2**	Marks Alpha RGCs (67+/5%). Functions as a co-marker of F-mini RGC	[68,69]
**Anti-Paralbumin (PV)**	Marks Alpha RGCs (73+/−4%)	[68]
**Anti-Melanopsin**	May works as co-marker of ON-sustained Alpha RGCs (M4) M1: soma, dendrites and axons up to the optic nerve M2: all cellular structures are marked M3: all cellular structures are marked M4: Soma and dendrites, only with tyramide signal amplification [41] M5: Weak perisomatic staining, greater percentage with tyramide signal amplification. M6: Weak soma staining, only with tyramide signal amplification.	[24,68,70,71,72]
**Anti-Brn3a/Pou4f1**	May works as co-marker of ON Alpha RGCs. May be excluded from the ipsilateral pathway Marks >95% BD, DRD4, Cdh6-RGCs, and CART+ RGC	[64,68,73,74]
**Anti-Brn3c/Pou4f1**	May work as co-marker of OFF-Transient Alpha RGCs	[68]
**Anti-Calbindin**	May work as co-marker of ON-Sustained Alpha RGCs	[68]
**Anti-Foxp1**	Marks all ON F-RGC	[69]
**Anti-Foxp2**	Marks all F-RGC	[69]
**Anti-Calretinine**	Marks S-BGC, M-BGC and B-BGC	[66]

## Data Availability

Not applicable.

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
