# Peer review of "Tools and Biomarkers for the Study of Retinal Ganglion Cell Degeneration"

_ijms, 2022, doi:10.3390/ijms23084287_

Round 1

Reviewer 1 Report

The authors discussed biomarkers for the study of retinal ganglion cell degeneration. Several specific comments need to be addressed.

Line 95-98: explanation needs to be provided.  

While oxidative stress was discussed, no discussion on ferroptosis (programmed cell death due to phospholipid peroxidation) Maher et al., 2020 DOI: 10.1016/j.chembiol.2020.10.010;  jakaria et al., 2021 https://onlinelibrary.wiley.com/doi/full/10.1111/jnc.15519. The authors should discuss and provide insight into ferroptosis about RGC degeneration focusing on glutathione, lipid and iron metabolism. The can find several recent findings on these topics. 

Line no 511: '' activation of autophagy can also be harmful'' -how? Activation of ferritinophagy, NCOA4-mediated autophagy, by which ferritin undergoes degradation, leading to neuronal cell death. The authors should discuss ferritinophagy in their context. 

The conclusion is missing. The authors should summarise their review and provide insight into limitations and future studies. 

Minor comments:

The manuscript still needs proofreading, especially for punctuation and expression. 

Reviewer 2 Report

The review manuscript submitted by Corral-Domenge, et al., interesting and useful for the development of various delivery approaches for the RGC based therapy for not only eye, but also for the neurodegenerative disorders. However, it lacks the novelty and describes the limited information on the topic of the subject selected. It needs to be revised majorly with some concerns.

But it majorly covers the non-apoptotic biomarkers and also mechanisms, techniques used for RGC cell death. Rewrite the title with the main discussed areas covered in the review.

The abstract entirely covers with CNS focus only. Why the authors not covered for ocular disorders, especially some of cancers and glaucoma like diseases also effectively involved by the RGC suppression. Explain?

The main focus of this review is “the specific biomarkers currently used that allow marking the different known subtypes of RGC”. Please write the non-specific biomarkers identified and used for the RGC involvement and diagnosis as well.

Write the references for Table 1 as well.

Reviewer 3 Report

The title of the text „Biomarkers for the Study of Retinal Ganglion Cell Degeneration” and its abstract promise a presentation of biomarkers for RGC degradation. The text presents three levels of material: RGC classification and susceptibility, with methods to study differentiation of these neural cells, especially labeling and related optical methods, and antibodies, then the neurodegeneration with various probes, and, finally, non-apoptotic biomarkers.

All parts, separately, are interesting and well prepared. Together – they lack introduction to the methods, with the character of analyses described. A reader expecting specific biomarkers meets histology and, later, fluorescent probes. The material in the begining is rather advanced and presented with very specific vocabulary, limiting general interest.

In my opinion, probes and biomarkers cannot be treated in the same way. Probes are like tools, whereas biomarkers are to be detected and analyzed (for example, using these tools). After reading this text, with many low molecular probes and internal chemicals, but also macromolecular probes and native proteins, I think a clear distinction should be introduced, especially with the current title. The phrase “certain fluorescent markers” – may indicate naturally occurring compounds (biomarkers), labeling reagents (probes) or results of labeling (developed biomarkers). Here the examples were CapQ and Annexin-A5, which are rather reagents-probes than typical biomarkers.

The caption for Figure 1 serves as a conclusion for the text, maybe the Authors consider using a regular paragraph, summarizing their work.

Advanced optical methods are described in paragraph 4.1, despite frequent use in earlier parts of the texts.

With various methods introduced, I would suggest specifying which are used/may be used in vivo, and which require sampling (such information is included in some places, however, a general statement would be helpful).

There are many reagents (labeling, fluorescence), listed by abbreviations and in some places – by chemical names, but no chemical structure is presented.
(please verify the names as they are incorrect or wrongly divided/copied: line 133, DiAsp (4-[4-didecylaminos- tyryl]-N-methyl-pyridinium iodide), (2-hydroxystilbene.4,4´-dicarboxamidine bis (methasulfonate)) and others)

In the fragment (line 377) after apoptosis evidence is listed, there appears a sentence: These compounds may be intrinsically fluorescent or may be labeled with a radioactive moiety. These molecules are N, N'-didansyl-L-cystine, NST-732 and 729, ML-9 and ML-10 [106].... The compounds mentioned here  are introduced at the beginning of the paragraph, without details, as associated with certain imaging techniques. For readers less familiar with molecular probes, this may be confusing.
In the next paragraph of this chapter (4.4) the characteristics of biomarker appears for the first time.

Phrases with unclear meaning:

line 136:  These dyes, once they get the cell membrane, are actively transported

line 225: Previously, many of these diseases were associated with certain causes. (what changed? what is the current association?)

Details:

Line 115: Reference is needed: In addition, certain fluorescent markers allow imaging of specifically diseased cells in vivo, like CapQ and Annexin-A5, which are valuable tools to study pathological mechanisms of neurological and retinal degeneration.

Line 237: unfinished? For a more exhaustive comparison of the different diagnostic imaging techniques of the retina[24].

Line 300: Reference is needed: As this is one of the initial steps of the apoptotic cascade, it constitutes a much earlier marker than others, such as DNA fragmentation detected by the terminal labeling of deoxynucleotidyl transferase dUTP Nick (TUNEL).

Style and format:

in vitro, in vivo – italics should be used

Line 123: However, the optimal pulse required, in such a small tissue, is much less [26],

Line 170: or the membrane marker CD90 also known Thy 1.2;
against this proteins

technics or techniques?

line 308: This technique has demonstrated useful in the investigation

line 316: has made it possible to

line 329: [92](Guo et al., 2007).

line 335: Its

line 431: sampling for in vivo o in vitro

line 433: green fluorescence when oxidize by hydrogen peroxide

Bip or BiP?

line 538: process called atg proteins, atg5 atg4 or atg12

line 544: death receptors follow by the concomitant

line 555 Mithocondrial

(Supplementary) Abbreviaions:

Round 2

Reviewer 1 Report

The manuscript could be considered for publication. Still, the manuscript deserves language proofreading. 

Reviewer 2 Report

No further comments.